# Effect on Benzoic Acid Production of Yoghurt Culture and the Temperatures of Storage and Milk Heat Treatment in Yoghurts from Cow, Goat and Sheep Milk

**DOI:** 10.3390/foods10071535

**Published:** 2021-07-02

**Authors:** Klára Bartáková, Lenka Vorlová, Sandra Dluhošová, Ivana Borkovcová, Šárka Bursová, Jan Pospíšil, Bohumíra Janštová

**Affiliations:** Department of Animal Origin Food and Gastronomic Sciences, University of Veterinary Sciences Brno, Palackého tř. 1946/1, CZ-61242 Brno, Czech Republic; vorloval@vfu.cz (L.V.); dluhosovas@vfu.cz (S.D.); borkovcovai@vfu.cz (I.B.); bursovas@vfu.cz (Š.B.); pospisilj@vfu.cz (J.P.); janstovab@vfu.cz (B.J.)

**Keywords:** caprine, ovine, bovine, fermented dairy, hippuric acid

## Abstract

Yoghurts from cow, goat and sheep milk were produced and stored under defined conditions to monitor the influence of various factors on the benzoic acid content as determined by Ultra High Performance Liquid Chromatography (UHPLC). The highest level of benzoic acid was found in sheep yoghurt (43.26 ± 5.11 mg kg^−1^) and the lowest in cow yoghurt (13.38 ± 3.56 mg kg^−1^), with goat yoghurt (21.31 ± 5.66 mg kg^−1^) falling in between. Benzoic acid content did not show statistically significant variation until the second and third weeks of storage, and the dynamics of this variation varied depending on the type of yoghurt. The yoghurt culture containing different strains of *Lactobacillus delbrueckii* subsp. *bulgaricus* and *Streptococcus thermophilus* also affected the contents of benzoic acid. Further, the different storage temperatures (2 and 8 °C) as well as the temperatures used to milk heat treatment before yoghurt production (80, 85 and 90 °C) affected the amount of benzoic acid in different types of yoghurts.

## 1. Introduction

Benzoic acid can act as an antimicrobial agent and protect against a broad spectrum of bacteria, yeasts and molds that cause food-borne diseases and food spoilage [1,2,3]. This is the basis for benzoic acid being used as a preservative. Moreover, benzoic acid also occurs naturally in several foods such as fruits, vegetables, spices and nuts. It is also found in low concentrations in dairy products as it is produced by micro-organisms during fermentation [4,5,6].

According to the US Food and Drug Administration, benzoic acid and its potassium and sodium salts are generally recognised as safe [7]. On the other hand, several adverse effects, such as metabolic acidosis, asthma, urticaria, hyperpnoea and convulsions have been observed in sensitive individuals even when present at low doses [8,9]. The Joint FAO/WHO (Food and Agriculture Organization of the United Nations/World Health Organization) Expert Committee on Food Additives evaluated and established 5 mg per kg of body weight as the acceptable maximum daily intake for benzoic acid and benzoate salt [6].

Benzoic acid formation in milk and dairy products has several different origins. Hippuric acid present in milk is converted to benzoic acid by fermenting bacteria, especially those present in starter culture. Another route could be by the auto-oxidation of benzaldehyde and phenylalanine degradation as seen with certain strains of lactic acid bacteria [3,10]. Benzoic acid can be also produced by *ß*-oxidation during the catabolism of fatty acids in bacterial cells [11]. In all these cases, benzoic acid can be considered a natural component of milk and milk products.

The amount of benzoic acid formed is affected by the type of micro-organism, the type of milk and the temperature of fermentation [6,10,12,13,14,15]. Strains that are known to produce benzoic acid in milk include *Lactococcus lactis*, *Lactobacillus casei*, *Streptococcus thermophilus*, *Lactobacillus helveticus*, *Escherichia coli* and *Pseudomonas fluorescens* [16]. A number of studies suggest that the level of benzoic acid depends primarily on the level of hippuric acid in raw milk [3,12,14]. The presence of hippuric acid and other carboxylic acids such as orotic or uric acid in milk is likely due to lactational cellular secretion, post-secretory metabolism in milk and/or transfer across the blood–milk barrier in the milk-producing animal [17].

In the European Union, the use of food additives, including preservatives, is restricted by EC Regulation no. 1333/2008 of the European Parliament and of the Council on food additives [18]. According to the list of food additives approved for use [19], all additives, including benzoic acid, are not permitted for use in butter and other unflavoured milk products (pasteurised and sterilised milk, pasteurised cream, fermented milk products not heat treated after fermentation and buttermilk). Given this regulation, benzoic acid in fermented dairy products can become a serious problem as it is impossible to distinguish between natural occurrence and possible addition. A detailed picture of natural benzoic acid formation in fermented dairy products may, therefore, allow the establishment of appropriate limits in different fermented dairy products, depending on the type of milk (cow, goat, sheep).

This study, therefore, set out to compare the content of benzoic acid in yoghurts from cow, goat and sheep milk produced in-house (in our dairy technology pilot plant) from milk geographically originating in one region, and comparing it with the proportion of hippuric acid in the milk used to make the yoghurts. We further compared these results with the amount of benzoic acid in yoghurts available commercially, because the knowledge of the natural benzoic acid content in yoghurts is crucial for distinguishing it from that which was exogenously added. Another goal of the work was to monitor the dynamics of benzoic acid content of the in-house yoghurts during storage and to determine the effect on the rate of benzoic acid formation of (a) the particular yoghurt culture used, (b) the storage temperature and (c) the temperature used for heat treating the milk.

## 2. Materials and Methods

### 2.1. Yoghurt Preparation

#### 2.1.1. In-House Yoghurts

The yoghurts were produced in our dairy technology pilot plant. Raw milk (cow, goat and sheep milk, collected in the summer from conventional farms in the Czech Republic) was heat treated using a discontinuous method with three temperature regimes (85, 90 and 95 °C for 10 min). Three types of commercial yoghurt starter cultures containing *Lactobacillus delbrueckii* subsp. *bulgaricus* and *Streptococcus thermophilus* were used, namely culture A (Milcom a.s., Prague, Czech Republic), culture B (Danisco, Paris, France) and culture C (Chr. Hansen, Hørsholm, Denmark), as these are the starters used in key dairies in the Czech Republic. All inoculated samples were incubated at 40 °C for 4–7 h with continuous pH measurement until they reached a pH of 4.5. The finished yoghurts were then stored for 3 weeks in two different temperature-controlled refrigerators with temperatures set at 2 and 8 °C. To determine the benzoic acid content, samples were taken from each batch starting the morning after yoghurt production and then repeated every 3–4 days. Seven samples were taken for each yoghurt batch during storage, as can be seen from the x-axis description in Figure 1. For each type of milk, 3 batches of yoghurts were produced, thus we analysed a total of 1134 yoghurt samples.

#### 2.1.2. Commercially Available Yoghurts

Yoghurts from cow (*n* = 37), goat (*n* = 26) and sheep (*n* = 22) milk were obtained either from shops in the Czech Republic or directly from individual cow, goat and sheep farms in the South Moravian Region during the summer season.

### 2.2. Analysis of Benzoic and Hippuric Acids

The content of benzoic and hippuric acids was determined using an Acquity UPLC system with UV detection (Waters, Milford, MA, USA). Preparation of the sample for UPLC determination was conducted according to the International Standard ISO 9231:2008(E) [20]. Fats and proteins contained in yoghurt were removed using precipitating solutions of potassium hexacyanoferrate(II) (Penta, Prague, Czech Republic) and zinc sulfate (Penta, Prague, Czech Republic) in a slightly alkalinized yoghurt solution (alkalization with sodium hydroxide (Penta, Prague, Czech Republic)). The precipitated yoghurt solution was then diluted with HPLC grade methanol (Merck, Darmstadt, Germany).

A UPLC BEH C18 chromatographic column (1.7 µm, 2.1 × 50 mm; Waters, Dublin, Ireland) was used for separation. The mobile phase was acetonitrile (HPLC grade; Merck, Darmstadt, Germany) and 0.05 M ammonium acetate (Sigma-Aldrich, Hamburg, Germany) in acetic acid (10:90, *v/v*; Penta, Prague, Czech Republic). Chromatography was performed in isocratic mode, the flow rate of the mobile phase was 0.5 mL min^−1^, the injection size was 1.4 µL, the column temperature was 35 °C and the detection wavelength was 227 nm. Data collection and evaluation using a calibration line was performed using Empower 2 software (Waters, Milford, MA, USA). The limit of detection was set at 3 times the S/N (signal/noise) ratio: 0.63 mg kg^−1^ for benzoic acid (Sigma-Aldrich, Hamburg, Germany) and 0.77 mg kg^−1^ for hippuric acid (Sigma-Aldrich, Hamburg, Germany). The determination limits (the lowest points of the calibration line) were 2.11 mg kg^−1^ for benzoic acid and 2.55 mg kg^−1^ for hippuric acid.

### 2.3. Statistical Analysis

The results were statistically evaluated using Statistica 13.2 software (Dell, Inc., Tulsa, OK, USA). Comparison of benzoic acid contents (from in-house yoghurts) and hippuric acid contents (in milk) was performed using a repeated one-way analysis of variance and Tukey’s HSD test. A *t*-test with Welch correction was used to compare the benzoic acid contents of in-house yoghurts and commercially available yoghurts. The dynamics of benzoic acid levels during storage was evaluated using a 4-factor analysis of variance and Tukey’s HSD test.

## 3. Results and Discussion

### 3.1. In-House Yoghurts: Content of Benzoic and Hippuric Acids

The content of benzoic acid in yoghurts made from cow, goat and sheep milk is given in Table 1, from which it is clear that the highest content was in sheep yoghurt (43.26 ± 5.11 mg kg^−1^) and the least in cow yoghurt (13.38 ± 3.56 mg kg^−1^). The differences in the amount of benzoic acid are statistically significant (*p* < 0.001) between the types of yoghurt (sheep, goat, cow). The results show that sheep yoghurt contains naturally higher levels of benzoic acid compared to that in goat and cow yoghurts, which corroborates the results of Horníčková et al. [14] who found that goat yoghurts (7.87 ± 2.21 mg kg^−1^) have a significantly (*p* < 0.001) lower amount of benzoic acid in sheep yoghurts (69.62 ± 26.09 mg kg^−1^). A statistically significant (*p* < 0.05) difference was also found in fermented goat drinks (20.3 ± 13.9 mg kg^−1^) than in fermented sheep drinks (29.5 ± 16.1 mg kg^−1^). This difference is probably due to the content of hippuric acid (see Table 1), which is a precursor of benzoic acid. We found significantly (*p* < 0.001) higher levels of hippuric acid in sheep milk (71.37 ± 15.06 mg kg^−1^) than in goat milk (21.31 ± 5.66 mg kg^−1^) or cow milk (13.38 ± 3.56 mg kg^−1^). A similar result was found by Horníčková et al. [14], who found 43.3 ± 12.3 mg kg^−1^ hippuric acid in sheep milk and 15.5 ± 8.3 mg kg^−1^ in goat milk.

Carpio et al. [21] determined a relatively wide range of hippuric acid concentrations in goat milk from Spain (17.75–188.96 mg L^−1^). The highest values significantly exceed the amount of hippuric acid in sheep milk found in our study or by Horníčková et al. [14], but the lowest values are comparable. Carpio et al. [21] state that the content of hippuric acid in milk is influenced by various factors, such as seasonality or farming method (organic vs. conventional). Significantly higher amounts of hippuric acid in goat milk were reported by Güler et al. [22], who determined hippuric acid contents in goat milk from two breeds of goats (Shami and Kilis) local to Turkey. Samples were taken each month from May to October during the lactation period. The results vary between 54.87 ± 3.86 mg L^−1^ and 406.08 ± 45.05 mg L^−1^. As with Carpio et al. [21], the higher values can be attributed to the different geographical origin of samples along with different goat nutrition. Horníčková et al. [14] determined the amount of hippuric acid in goat and sheep milk during the lactation period from April to October. The hippuric acid content varied between 5.78 and 25.11 mg kg^−1^ in goat milk and between 20.29 and 55.58 mg kg^−1^ in sheep milk. Nevertheless, they did not observe any increasing or decreasing trend, as observed by Güler et al. [22].

Several authors [15,16,23,24,25,26,27,28] report benzoic acid contents in cow milk comparable to our results (5.29–20.72 mg kg^−1^), as is clear from the values from the published literature listed in Table 2. Comparing benzoic acid in yoghurts made from cow’s milk of different provenance and different geographical origins, it is clear that the variation in benzoic acid content does not depend on the breed, nutrition or management of farms in different geographical regions. This kind of comparison for goat and sheep yoghurts is not possible as there is not enough available published literature on such determinations.

### 3.2. Comparison of Benzoic Acid Content in Yoghurts Produced In-House and in Commercially Available Yoghurts

Table 3 shows the amount of benzoic acid in commercially available yoghurts. There was no statistically significant difference (*p* > 0.05) in the levels of benzoic acid in cow, goat and sheep yoghurts. From the minimum and maximum values of the determined concentrations of benzoic acid in commercially available yoghurts, it is clear that the range of benzoic acid concentrations in all three types of yoghurts was very similar (3.25–54.36 mg kg^−1^ in cow yoghurts, 4.87–40.94 mg kg^−1^ in goat yoghurts and 11.37–51.19 mg kg^−1^ in sheep yoghurts).

Comparing the concentration of benzoic acid in yoghurts produced in-house (Table 1) and in commercially available yoghurts (Table 3), we found that the amount of benzoic acid in cow and goat yoghurts (both commercially available and in-house) did not differ significantly (*p* > 0.05). However, in several commercially available yoghurts, the concentration significantly exceeded the values found in other samples. In the case of cow yoghurts, the level of benzoic acid was more than twice the maximum levels detected in the in-house samples. This raises the question of unauthorized addition of benzoic acid. The situation was different for yoghurts made from sheep milk: the benzoic acid content in commercially available samples was statistically significantly lower than the in-house samples (*p* < 0.01). This indicates the possible addition of cow’s milk in the production of sheep yoghurt, because we found a lower rate of benzoic acid formation in cow yoghurt than in sheep yoghurt. The addition of a different type of milk in the manufacture of yoghurt is allowed if the concentration of sheep milk in producing sheep yoghurt is more than 50%. Nonetheless, this addition must be indicated on the packaging of the yoghurt. However, the packaging on the analysed yoghurts contained no indication of any addition of milk other than sheep milk.

Horníčková et al. [14] determined the amount of benzoic acid in yoghurts produced on a farm that bred goats and sheep. They found a lower level of benzoic acid (7.87 ± 2.21 mg kg^−1^) in goat yoghurts than we found (both in samples from commercially available as well as in-house yoghurts). In contrast, in sheep yoghurts, we found a significantly higher average level of benzoic acid (69.62 ± 26.09 mg kg^−1^) than the maximum we determined in our samples from commercially available as well as in-house yoghurts.

### 3.3. Yoghurt Storage

We found statistically significant changes in the content of benzoic acid over the 21 days of storage. Figure 1 shows changes in the average benzoic acid content in cow, goat and sheep yoghurts during storage. It is clear that there were no statistically significant changes in the amount of benzoic acid in any type of yoghurt in the first week of storage, but over the following 2 weeks there were distinct changes. On the 11th day of storage, we found a statistically significant (*p* < 0.01) decrease in benzoic acid content in cow yoghurts to a value that was the lowest over the 3-week duration. On the other hand, in goat and sheep yoghurts, we found a statistically significant (*p* < 0.01) increase, with goat yoghurts reaching a maximum on the 11th day and sheep yoghurts on the 14th day of storage. For cow yoghurts, the maximum benzoic acid content was recorded on the 18th day of storage.

**Figure 1 foods-10-01535-f001:**
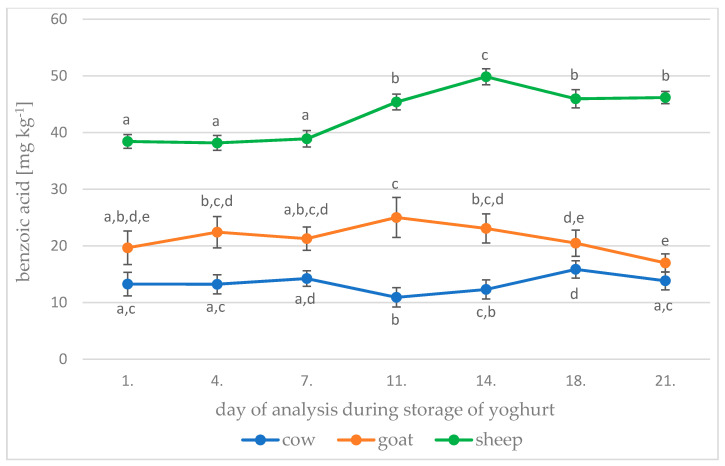
Changes in benzoic acid content in cow, goat and sheep yoghurts during storage. Different letters (a–e) for each type of yoghurt indicate a statistically significant difference (*p* < 0.01).

Garmiene et al. [16] found similar results during storage of fermented cow’s milk for 5 and 10 days at 5–6 °C, with increased benzoic acid content in cow’s milk fermented by *Lactobacillus delbrueckii* subsp. *bulgaricus* (increase from 13.18 ± 0.18 mg kg^−1^ to 13.80 ± 0.15 mg kg^−1^) on the fifth day of storage. They found that it subsequently decreased to 13.55 ± 0.21 mg kg^−1^ on the 10th day of storage.

A different result is reported by Urbienė and Leskauskaitė [27], who stored cow yoghurts for 6 days at 6 ± 1 °C and observed a gradual statistically significant decrease in benzoic acid levels from the initial 17, 19 and 25 mg kg^−1^ for three different yoghurt cultures to 9, 11 and 14 mg kg^−1^.

### 3.4. Factors Influencing Benzoic Acid Content

#### 3.4.1. Yoghurt Culture

Figure 2 shows the effect of the yoghurt culture used on the production of benzoic acid. The influence of yoghurt culture was different in each type of yoghurt. In cow yoghurts inoculated with yoghurt culture A, the amount of benzoic acid (16.19 ± 0.71 mg kg^−1^) was significantly higher (*p* < 0.01) than in cow yoghurts inoculated either with culture B (11.99 ± 0.92 mg kg^−1^) or culture C (11.98 ± 1.06 mg kg^−1^).

Similarly, significantly (*p* < 0.01) more benzoic acid was found in goat yoghurts inoculated with culture A (22.46 ± 1.42 mg kg^−1^) compared to yoghurts inoculated with culture B (19.18 ± 2.40 mg kg^−1^). However, in contrast to cow yoghurts, the amount of benzoic acid in goat yoghurts inoculated with culture C (22.14 ± 1.59 mg kg^−1^) was significantly (*p* < 0.01) higher than in yoghurts inoculated with culture B.

In sheep yoghurts, a significantly (*p* < 0.01) lower level of benzoic acid was found in yoghurts inoculated with yoghurt culture C (42.11 ± 1.91 mg kg^−1^) compared to that in yoghurts inoculated either with culture A (43.55 ± 1.70 mg kg^−1^) or culture B (44.11 ± 1.43 mg kg^−1^).

From the above results, it is clear that the choice of yoghurt culture can significantly affect the amount of benzoic acid in the final product, and this conclusion is confirmed by further studies [15,16,27]. For instance, Garmiene et al. [16] found that the amount of benzoic acid in cow’s milk inoculated with strains 21 and 148/3 of *Lactobacillus delbrueckii* subsp. *bulgaricus* is 4.5 times lower than in milk fermented by strain 14. Yu et al. [15] found statistically significant variations in the amount of benzoic acid in cow yoghurts produced using various strains of *Lactobacillus* spp.

On the other hand, Horníčková et al. [14] did not find a statistically significant difference in the amount of benzoic acid in goat or sheep milk fermented by different cultures.

#### 3.4.2. Storage Temperature

Figure 2 shows the effect of varying storage temperatures on the rate of benzoic acid production in the three types of yoghurts. In cow yoghurts, a statistically significantly higher (*p* < 0.01) rate of benzoic acid production was evident in yoghurts stored at 8 °C (13.99 ± 0.86 mg kg^−1^) than in yoghurts stored at 2 °C (12.77 ± 0.90 mg kg^−1^). In goat and sheep yoghurts, the storage temperature had no statistically significant effect on the rate of benzoic acid production. Nonetheless, it is clear that the amount of benzoic acid was slightly higher in yoghurt stored at 8 °C than in yoghurt stored at 2 °C. This result suggests that at 8 °C, there was basal-level benzoic acid production, which was not seen at 2 °C. Similarly, Garmiene et al. [16] observed a small increase in benzoic acid content in yoghurts stored at 5–6 °C.

#### 3.4.3. Heat Treatment Temperature

As with storage temperature, the temperature used to heat treating the milk from which the yoghurts were made affects the rate of benzoic acid production (Figure 2). In cow yoghurts, milk heat treated at 90 °C yielded the highest amount of benzoic acid. The amount of benzoic acid in these yoghurts was significantly (*p* < 0.01) higher than in yoghurts made from cow’s milk heat treated at 85 °C or 95 °C. In goat yoghurts, milk heat treated at 85 °C yielded the lowest amount of benzoic acid; the amount of benzoic acid in yoghurts made from milk heat treated at 90 °C or 95 °C did not differ significantly. No statistically significant difference was found in the amount of benzoic acid in any of the three yoghurts made from sheep milk heat treated at 85 °C, 90 °C and 95 °C. From the goat yoghurt results, it can be presumed that at the higher milk treatment temperature, benzoic acid formation occurs via a different pathway, namely through phenylalanine degradation (as reported by Hejtmánková et al. [10]), because the phenylalanine content of goat milk is the highest among the three types of milk [29].

## 4. Conclusions and Future Perspective

In this study, we produced yoghurt from cow, goat and sheep milk, and stored them under defined conditions. We found the highest amount of benzoic acid in sheep yoghurts and the lowest amount in cow yoghurts, which is exactly in line with the content of hippuric acid in the milk from which the yoghurts were made. In samples of commercially available yoghurts, the average content of benzoic acid in cow and goat yoghurts was found to be comparable to the amount of benzoic acid in the yoghurts produced in-house, but the amount of benzoic acid was significantly (*p* < 0.01) lower in commercially available sheep yoghurts than those produced in-house. We also found benzoic acid at levels significantly (*p* < 0.01) higher than naturally occurring concentrations in samples of commercially available cow and goat yoghurts.

We also followed changes in benzoic acid content over 21 days of storage, and there were no statistically significant changes in any of the three types of yoghurts during the first week. Over the next 2 weeks, the content of benzoic acid varied with different dynamics in each of the different types of yoghurt. We also found that the yoghurt culture, the storage temperature of yoghurts and the temperature used for heat treatment of milk prior to yoghurt production can influence the amount of benzoic acid.

Our results have been submitted to the Czech regulatory authorities because this study was conducted following their request to determine the range of naturally occurring benzoic acid in different types of yoghurt. It follows that the content of benzoic acid can be influenced by a number of factors, and these must be taken into account when setting regulatory limits on the range of naturally occurring benzoic acid in individual types of yoghurts. It is, therefore, important to study further the influence of various factors on benzoic acid content, in particular in yoghurts from goat and sheep milk. It may also be useful to focus on yoghurt production using the probiotic *Streptococcus thermophilus* strain and to compare the formation of benzoic acid between this type of yoghurt and conventional yoghurts.

## Figures and Tables

**Figure 2 foods-10-01535-f002:**
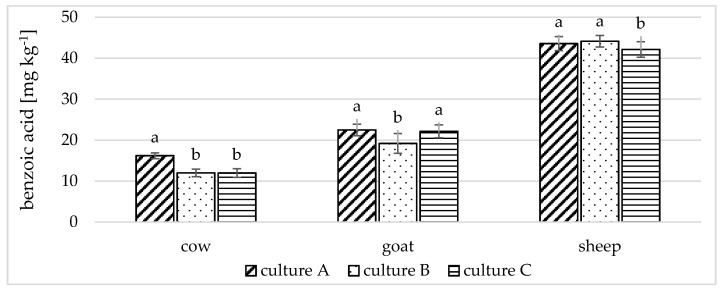
The amount of benzoic acid in cow, goat and sheep yoghurts inoculated with three different cultures (A, B, C); stored at two different temperatures (2 and 8 °C) and made from milk heat treated at different temperatures (85, 90, 95 °C). Different letters (a, b) for each type of yoghurt indicate a statistically significant difference (*p* < 0.01).

**Table 1 foods-10-01535-t001:** Benzoic acid content of in-house cow, goat and sheep yoghurts and hippuric acid content in cow, goat and sheep milk used for the production of yoghurts.

Type of Yoghurt/Milk	Benzoic Acid [mg kg^−1^]			Hippuric Acid [mg kg^−1^]		
	Mean ± SD	Min.	Max.	Mean ± SD	Min.	Max.
Cow	13.38 ± 3.56 ^a^	5.29	20.72	16.17 ± 1.94 ^a^	13.96	17.58
Goat	21.31 ± 5.66 ^b^	5.43	30.72	35.10 ± 2.87 ^b^	31.71	40.32
Sheep	43.26 ± 5.11 ^c^	31.60	54.34	71.37 ± 15.06 ^c^	53.86	98.58

Different letters (a–c) in the same column indicate a statistically significant difference (*p* < 0.001).

**Table 2 foods-10-01535-t002:** Published results of benzoic acid in yoghurts from cow’s milk.

Benzoic Acid [mg kg^−1^]		Country of Origin	Reference
Mean ± SD	Range		
	8.94–28.30	Turkey	[23]
20.09 ± 2.81		Turkey	[24]
24.48 ± 8.50		Turkey	[25]
	0.77–17.46	Korea	[15]
	0–16.5	Korea	[26]
15.28 ± 1.06		Lithuania	[16]
	16–24	Lithuania	[27]
	1.5–5.0	Iran	[28]

**Table 3 foods-10-01535-t003:** Benzoic acid content in commercially available cow, goat and sheep yoghurts.

Type of Yoghurt	Benzoic Acid [mg kg^−1^]		
	Mean ± SD	Min.	Max.
Cow	18.65 ± 15.21 ^a^	3.25	54.36
Goat	22.45 ± 10.87 ^a^	4.87	40.94
Sheep	23.13 ± 12.18 ^a^	11.37	51.19

The same letters (a) indicate no statistically significant difference (*p* > 0.05).

## Data Availability

Data is available upon request to the authors.

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
