# Peer review of "Effect on Benzoic Acid Production of Yoghurt Culture and the Temperatures of Storage and Milk Heat Treatment in Yoghurts from Cow, Goat and Sheep Milk"

_foods, 2021, doi:10.3390/foods10071535_

Round 1
Reviewer 1 Report
Dear authors,
The manuscript is worthy, I indicate some suggestions, I hope useful to improve it.
In introduction, line 82. "In our technology pilot plant". You adopt the same sentence in Materials and Methods, at line 94. In my opinion is a redundance, please, adopt a different phrase, moreover I suggest a limited use of the possessive pronoun “our” in a scientific article. At line 84 the word “we” shows the same cruciality.
In results and discussion, lines 160-167. “The highest content was in sheep yoghurt (43.26 ± 5.11 mg kg-1) and the least in cow yoghurt (13.38 ± 3.56 mg kg-1). The differences in the amount of benzoic acid are statistically significant (P < 0.001) between the types of yoghurt (sheep, goat, cow). The results show that sheep yoghurt contains naturally higher levels of benzoic acid compared to that in goat and cow yoghurts”: In italic: this is a redundance, you still described this result at the beginning of the sentence. Improve the phrase please.
In results and discussion, lines 167-168. Please can you adopt a better sentence? Obviously you reported milk-based drinks but in my opinion should be indicated (goat drinks, sheep drinks!). Thanks.
In conclusions, lines 549 and 723. In the sentence you adopted the word “we”. In the conclusions I think you can adopt the first person, useful to highlight the results.
Bibliography. Very few sources. It is possible?
Author Response
Dear reviewer,
Our team of authors wish to express our thanks for taking the time to conduct a review of our manuscript entitled "Effect on benzoic acid production of yoghurt culture and the temperatures of storage and milk heat treatment in yoghurts from cow, goat and sheep milk", as well as for your insightful comments, questions and recommendations. Following your suggestions, we have modified the manuscript and highlighted the changes in MS Word using the "Track Changes" function.
Attached answers to individual comments:
(We don't know why, but on our computers, line numbering is different than what you type. Therefore, in our responses to your individual comments, we use line numbers according to our computers.)
- In introduction, line 82. "In our technology pilot plant". You adopt the same sentence in Materials and Methods, at line 94. In my opinion is a redundance, please, adopt a different phrase, moreover I suggest a limited use of the possessive pronoun “our” in a scientific article. At line 84 the word “we” shows the same cruciality.
We removed the possessive pronoun "our" (on my computer these lines are marked with the numbers 70 and 83) and the sentence with the subject "we" was transferred to the passivum (on lines 72-73).
- In results and discussion, lines 160-167. “The highest content was in sheep yoghurt (43.26 ± 5.11 mg kg-1) and the least in cow yoghurt (13.38 ± 3.56 mg kg-1). The differences in the amount of benzoic acid are statistically significant (P < 0.001) between the types of yoghurt (sheep, goat, cow). The results show that sheep yoghurt contains naturally higher levels of benzoic acid compared to that in goat and cow yoghurts”: In italic: this is a redundance, you still described this result at the beginning of the sentence. Improve the phrase please.
We removed the redundant part of the sentence (on lines 138-140).
- In results and discussion, lines 167-168. Please can you adopt a better sentence? Obviously you reported milk-based drinks but in my opinion should be indicated (goat drinks, sheep drinks!). Thanks.
We have added the terms "milk-based" to goat and sheep drinks to make it clear (on lines 144-145).
- In conclusions, lines 549 and 723. In the sentence you adopted the word “we”. In the conclusions I think you can adopt the first person, useful to highlight the results.
Thank you!
- Very few sources. It is possible?
We used all available literary sources on the topic. In our opinion, the lower number of sources indicates the need to solve this problem.
We hope that the modifications we made satisfy your requirements.
Yours sincerely Klára Bartáková
Reviewer 2 Report
Bartáková et al investigated yoghurts from cow, goat and sheep milk in order to to monitor the influence of various factors on the benzoic acid content as determined by UHPLC. The highest level of benzoic acid was found in sheep yoghurt and the lowest in cow yoghurt. The yoghurt culture containing different strains of Lactobacillus delbrueckii subsp. bulgaricus and Streptococcus thermophilus also affected the contents of benzoic acid. Moreover, the different storage temperatures as well as the temperatures used to milk heat treatment before yoghurt production affected the amount of benzoic acid in different types of yoghurts. Moreover this article is already revised by authors based on reviewer(s) recommendations. The article is nicely written. after following minor revision.
[1] Reference section should be arranged according to Journal format
Author Response
Dear reviewer,
Our team of authors wish to express our thanks for taking the time to conduct a review of our manuscript entitled "Effect on benzoic acid production of yoghurt culture and the temperatures of storage and milk heat treatment in yoghurts from cow, goat and sheep milk", as well as for your encouraging comment. Following your suggestion, we corrected some inaccuracies in the references section and highlighted the changes in MS Word using the "Track Changes" function (lines 409, 414-417, 421, 428, 434-436, 455-456, 459, 464-466).
We hope that the modifications we made satisfy your requirements.
Yours sincerely Klára Bartáková
This manuscript is a resubmission of an earlier submission. The following is a list of the peer review reports and author responses from that submission.
Round 1
Reviewer 1 Report
Overview
The paper deals with monitoring the influence of various factors on the benzoic acid contents in commercial and handmade yoghurts from cow, goat and sheep milk as determined by Ultra Performance Liquid Chromatography (UHPLC). The highest levels of benzoic acid (43.3±5.11 mg kg-1) were found in sheep’s milk yoghurt, and the lowest (13.4±3.56 mg kg-1) in cow’s milk yoghurt. Contents of benzoic acid did not show statistically significant differences until the second and third weeks of storage, and the dynamics of the deviations varied depending on the type of yoghurt, the yoghurt culture used (different strains of Lactobacillus delbrueckii susp. bulgaricus and Streptococcus thermophilus), the different storage temperatures (2 and 8 ºC) and the temperatures used in the heat treatment of the yoghurt milk.
In my opinion, there are serious deficiencies both in relation to the approach and in regard to formal aspects of the study. The experimental design seems confusing in relation to the sampling of the yoghurts, as it is not specified how many samples were analysed. The heat treatment applied to the milk for the manufacture of artisanal yoghurts is not a (HTST) pasteurisation treatment, as described in the manuscript. The reasons for the differences attributable to the various factors analyzed, such as the storage temperature or the milk heating temperature have not been discussed or explained. The amounts of benzoic acid determined are relatively low from a health point of view, and could be primarily related to the levels of hippuric acid in the milk as stated in the Introduction section. From a formal point of view, the text could have been shortened to a great extent, since the Results and discussion section includes some information that is not relevant to the study. The number of figures is also excessive. Finally, the text is poorly constructed in numerous sentences and should have been proofread by a native English speaker.
General comments
- The English language should have been revised by a native English speaker with some knowledge of the subject.
- The Results and discussion and the Conclusion sections should be shortened.
- The number of figures should be reduced.
Detailed comments
Title
- Change to e.g. “…storage and milk heat treatment…”.
Abstract
- p. 1. Line 13. Change to “Ultra Performance Liquid Chromatography (UHPLC)”.
- p. 1. Line 17–19. Change to e.g. “The yoghurt culture containing different strains of… also affected the contents of benzoic acid.”
- p. 1. Lines 21–22. Delete the last sentence of the abstract, as it does not provide any information about the results of the study.
Introduction
- p. 1. Line 29. Change to “spices and nuts”.
- p. 1. Line 41. Change to e.g. “Another routes could be…”
- p. 2. Line 67. Change to e.g. “dairy tecnhnology pilot plant”. Also on Line 79.
- p. 2. Lines 70–71. Change to e.g. “because the knowledge of the natural benzoic acid content”.
Materials and Methods
- p. 2. Line 81. In the dairy industry, high temperature pasteurisation (HTST treatments) is not applied for periods of minutes. Plate heat exchangers are generally used, and holding periods do not usually exceed 30 seconds (maximum 1 minute). The only treatment that is applied "discontinuously" is the LTLT pasteurisation in vat (60-65 ºC for 30 minutes).
- p. 2. Line 87. Change “thermostats” to e.g. “refrigerators”. A thermostat is a device used to control temperature.
- p. 2. Lines 89–90. There is no rigor in: “starting the morning after… repeated every 3-4 days”. How many “in-house” yoghurts were sampled?
- p. 3. Lines 92–94. How many commercial yoghurts were collected?
- p. 3. Line 97. Provide the State for Waters.
- p. 3. Line 104. Is Waters manufacturer addressed in Ireland?
- p. 3. Line 111. Is “S/N” the Signal-to-Noise ratio?
Results and Discussion
- p. 3. Line 126. Check for “from which it is clear”.
- p. 3. Lines 127–128. Delete the sentence “Goat yoghurt fell…”. It is not necessary to repeat in the text all the results shown in tables.
- p. 3. Line 129. Delete “highly”. Better write “P” instead of “p”, here and throughout the manuscript.
- p. 3. Lines 131–132. Do not repeat all the results in the text.
- p. 3. Lines 134–136. Delete “This”. Delete “in fermented drinks with a lower level of benzoic acid”. The amounts of benzoic acid are relatively low considering the FAO recommendations (maximum daily intake of 5 mg per kg of body weight).
- p. 4. Lines 147–158. Shorten the paragraph, as most of the information is not relevant to the study.
- p. 4. Line 148. Check for “While the lowest” at the beginning of the sentence.
- p. 4. Lines 154–158. Change “l” to “L”.
- p. 4. Line 176. Table 2. Delete “of Values”.
- p. 5. Line 193. Delete “for commercially available yoghurts”. Rewrite.
- p. 5. Line 197. Avoid using long dashes, also on Line 208.
- p. 5. Line 207. Change to e.g. “bred goats and sheep”.
- p. 5. Lines 208–209. Change to e.g. “than that found in the present study”. Change to “in sheep yoghurts”. Rewrite.
- pp. 6 and 7. Try to reduce the three figures (1, 2 and 3) to just one.
- p. 7. Line 236–254. Shorten the paragraphs. Starter cultures other than yoghurt cultures are used in the studies mentioned. It is not necessary to transcribe all numerical results.
- pp. 8–10. Try to reduce the three figures (4, 5 and 6) to just one.
- p. 8. Lines 267–277. Check for e.g. “significantly (p <0.01) more benzoic acid was found”. Rewrite the paragraphs. It is not necessary to describe the contents in detail. On the other hand, the differences do not seem relevant enough.
- p. 8. Lines 278–294. Shorten the paragraph. Do not provide so many details and numerical results. Fermented milks that are made with microorganisms other than Lactobacillus delbrueckii susp. bulgaricus and Streptococcus thermophilus are not yoghurts.
- p. 8. Lines 297–302. Shorten the paragraph. Do not provide so many numerical results.
- p. 9. Lines 304–311. Try to discuss or argue the results, not just describing them. What could the differences be attributed to? Delete the last sentence.
- pp. 9 and 10. Lines 316–317. The yoghurt milks were heated to different temperatures, but were not strictly pasteurised (HTST treatment). Try to discuss or explain the results, not just describing them. Delete the last sentence.
Conclusions
- p. 10. Lines 332–356. Shorten the section. Most of the information presented are not conclusions of the study. Some comments are quite speculative (would it really make sense to add benzoic acid to a shelf-stable product like yoghurt?). Check for: “Our results are can thus directly inform…” The reasons for the differences attributable to the various factors analysed, such as the storage temperature or the milk heating temperature, have not been discussed or argued for.
Reviewer 2 Report
Dear authors,
I read your manuscript. The comments could be useful to improve an interesting work.
In abstract. At line 13. As often happens, in the scientific articles several acronyms are adopted, however, although some of them are very common, is useful to indicate, at the first time, the name in full. For example UHPLC as Ultra High Performance Liquid Chromatography, or FAO (line 36, in introduction) as Food and Agriculture Organization of the United Nations.
In abstract. At line 73. In the in house yoghurts. I think that you can to improve this construction (dissonance). I read a better sentence at lines 119-120 “..the benzoic acid contents of in-house yoghurts and commercially available yoghurts”.
In materials and methods. At line 98. International Standard ISO 9231:2008. Please can you check this source? Maybe you should update it.
In material and methods. As described below, this section must be improved. In results are described the samples, this is a typical topic of materials and methods. Moreover, please, can you give more accuracy in the description of the samples? Perhaps I’m wrong though I didn't see precise numbers (e.g. samples of goat milk taken in winter – February… how many samples? Every day? Weekly?) You compare milk of cow, goat and sheep, a better description is required. Thanks a lot.
In results and discussion. At line 155 you often reported, described as results, topics that would be to be inserted in materials and methods.
In results and discussion. At line 199 An interesting matter is introduced: “This indicates the possible addition of cow’s milk in the production of sheep yoghurt, because we found a lower rate of benzoic acid formation in cow yoghurt than in sheep yoghurt”. If this sentence is not well argued, the important argument (food fraud) may look like a off topic theme.
Sincerely
All the best

Reviewer 3 Report
In my opinion this study is very important to food safety. Results yours study can be true and interesting for screenig. I can not found any mistakes. Congratulations.
Reviewer 4 Report
Bartáková et al demonstrated that yoghurts from cow, goat and sheep milk were produced and stored under defined conditions to monitor the influence of various factors on the benzoic acid content. The highest level of benzoic acid was found in sheep yoghurt, and the lowest in cow yoghurt. Benzoic acid content did not show statistically significant variation until the second and third weeks of storage, and the dynamics of this variation varied depending on the type of yoghurt. They used three types of yoghurt culture containing Lactobacillus delbrueckii and Streptococcus thermophilus. They illustrated that benzoic acid at levelssignificantly higher than naturally occurring concentrations in samples of commercially available cow and goat yoghurts and this suggests unauthorized addition of benzoic acid. The study look very interesting in food perspective and can be accepted after following major revision.
[1] Some other studies have been reported regarding benzoic acid and yogurt and authors mention in introduction that why this study is important and how this different from previous studies. One missing ref.(i)
[i] Hejtmánková A., Horák V., Dolejšková J., Louda F., Dragounová H. (2000): Influence of yogurt cultures on benzoic acid content in yoghurt. Czech J. Food Sci., 18: 52-54.
[2] Conclusion part should be change to “Conclusion and future perspective” and author should mention few sentences about the future directions.